# Wood Extractives of Silver Fir and Their Antioxidant and Antifungal Properties

**DOI:** 10.3390/molecules26216412

**Published:** 2021-10-23

**Authors:** Viljem Vek, Eli Keržič, Ida Poljanšek, Patrik Eklund, Miha Humar, Primož Oven

**Affiliations:** 1Department of Wood Science and Technology, Biotechnical Faculty, University of Ljubljana, Jamnikarjeva 101, SI-1000 Ljubljana, Slovenia; eli.kerzic@bf.uni-lj.si (E.K.); ida.poljansek@bf.uni-lj.si (I.P.); Miha.Humar@bf.uni-lj.si (M.H.); 2Johan Gadolin Process Chemistry Centre, Åbo Akademi University, FI-20500 Turku, Finland; patrik.c.eklund@abo.fi

**Keywords:** *Abies alba*, knotwood, chromatography, lignans, antioxidant properties, antifungal properties

## Abstract

The chemical composition of extractives in the sapwood (SW), heartwood (HW), knotwood (KW), and branchwood (BW of silver fir (*Abies alba* Mill.) was analyzed, and their antifungal and antioxidant properties were studied. In addition, the variability of extractives content in a centripetal direction, i.e., from the periphery of the stem towards the pith, was investigated. The extracts were analyzed chemically with gravimetry, spectrophotometry, and chromatography. The antifungal and antioxidative properties of the extracts were evaluated by the agar well diffusion method and the diphenyl picrylhydrazyl radical scavenging method. Average amounts of hydrophilic extractives were higher in KW (up to 210.4 mg/g) and BW (148.6 mg/g) than in HW (34.1 mg/g) and SW (14.8 mg/g). Extractives identified included lignans (isolariciresinol, lariciresinol, secoisolariciresinol, pinoresinol, matairesinol) phenolic acids (homovanillic acid, coumaric acid, ferulic acid), and flavonoids epicatechin, taxifolin, quercetin). Secoisolariciresinol was confirmed to be the predominant compound in the KW (29.8 mg/g) and BW (37.6 mg/g) extracts. The largest amount of phenolic compounds was extracted from parts of knots (281.7 mg/g) embedded in the sapwood and from parts of branches (258.9 mg/g) adjacent to the stem. HW contained more lignans in its older sections. Hydrophilic extracts from knots and branches inhibited the growth of wood-decaying fungi and molds. KW and BW extracts were better free radical scavengers than HW extracts. The results of the biological activity tests suggest that the protective function of phenolic extracts in silver fir wood can also be explained by their antioxidative properties. The results of this study describe BW as a potential source of phenolic extractives in silver fir.

## 1. Introduction

Recently, woody biomass has attracted much scientific attention in which it is treated as a potential source of bioactive phytochemicals, also known as extractives [1,2,3]. These natural compounds are not equally distributed in tree tissues. Some parts of trees, e.g., knots and bark, have been reported to be a rich and cheap natural source of polyphenols [1,2,4]. Silver fir (*Abies alba* Mill.) is an evergreen conifer that is one of the most valuable tree species in Europe, for historical and economic reasons. The importance of silver fir is based on its shade tolerance, ability to adjust to environmental conditions and to coexist with many tree species, which enables maintaining high biodiversity in forest ecosystems [5].

Silver fir knots are considered to be the richest source of lignans. Willför et al. [1,2,6] comprehensively investigated the contents of hydrophilic extractives and individual lignans that can be extracted from knotwood (KW) and stemwood (SW) of various conifer species. Knots of silver fir contained significantly higher amounts of hydrophilic extractives than SW or HW. The difference in hydrophilic extractives content between KW samples of living and dead branches was not significant. On the other hand, living knots (LK) contained greater amounts of sugars and sugar alcohols compared to dead knots (DK) [2]. The lignan secoisolariciresinol was the dominant compound in KW extracts, while matairesinol was characteristic of the HE samples. HE contained significantly lower concentrations of all lignans than did knots. Chromatographic analysis showed that the KW extracts contained lariciresinol, 7-hydroxymatairesinol, 7-allo-hydroxymatairesinol, liovil, matairesinol, cyclolariciresinol, pinoresinol, and nortrachelogenin, in addition to secoisolariciresinol [2]. It has also been reported that silver fir is the only *Abies* species containing the mono- and dimethyl ethers of secoisolariciresinol. Larger substances, i.e., sesquineolignans, dineolignans, and higher oligolignans, have been found in all KW samples of silver fir. HW and KW have also been reported to contain sugars and sugar alcohols [2,7]. Tavčar Benković et al. [8] reported that aqueous extract from the wood of silver fir branches contains lignans, which constitute about 10% of the extract and include isolariciresinol, hydroxymatairesinol, secoisolariciresinol, lariciresinol, pinoresinol, and matairesinol. Despite large amounts of hydrophilic extractives in coniferous KW, these tissues also contain relevant amounts of lipophilic extractives [2,9]. However, the amount of lipophilic extractives is reported to be lower than the amount of hydrophilic extractives in coniferous knots [9]. As reported by Kebbi-Benkeder et al. [7], the hexane extracts of silver fir KW contained mainly terpenes, i.e., dehydrojuvabione and juvabione; these two were also referred to as dominant compounds, together with epimanool, tumerone, farnesol, eudesmol, nerolidol, and isopimaric acid. 

The content of phenolic extractives in the wood of a living tree is known to vary in both radial and longitudinal directions [10,11]. Kebbi-Benkeder et al. [7] aimed to design a model of the vertical distribution of extractive concentration of silver fir KW as a prerequisite for further studies on eco-physiological studies. They showed that the extractive concentration increases from the tree tip to the living crown base and from the tight to the loose knot parts. Accordingly, trees with the longest crowns that dominate or/and grow in thinned stands exhibit the highest knot extractive concentrations. Just recently the same group proved a significant longitudinal variation in the contents of ethanol-soluble extractives in the knots of a silver fir tree [12]. The highest concentrations of the extractives were found in knots located beneath the crown and in knots from the base of the crown. The largest concentrations of secoisolariciresinol, i.e., the most abundant lignan of silver fir, was found in knots taken at the transitional zone around the base of the crown [12].

A literature review shows that the chemical composition of silver fir wood extractives has already been extensively studied [2,7,12]. However, information on the content variability of extractives in stemwood and KW in a radial direction, as well as in BW, still remains unclear. For silver fir (*A. alba*), Hamada et al. [13] found the largest amounts of toluene/ethanol soluble extractives in juvenile and inner HE, with lower amounts measured for outer HW and SW samples. It is evident that the existing information on the variability of hydrophilic/phenolic extractives in the radial direction for stemwood and KW of silver fir is not clearly presented.

Phenolic compounds play an important role in the defense and protective mechanisms that occur in the wood of trees [7,14]. In general, polyphenols have three defensive functions: as biocides, radical scavengers, and metal chelators [15]. With their antifungal, antimicrobial, and antioxidant properties, phenolic extractives are treated as phytochemicals with potential applications, e.g., as bioactive agents in formulations for wood preservation, food protection, or even therapeutic purposes [3,16,17,18,19]. Lignans have been reported as compounds with high antioxidative potency and radical scavenging capacity [20]. Tavčar Benković et al. [8] also discovered that BW extract has greater antioxidant activity than ascorbic acid, resveratrol, or butylated hydroxytoluene (BHT) and similar antioxidant activity to epigallocatechin gallate. In addition, Pietarinen et al. [15] concluded that KW is a rich source of natural antioxidants, since knot extracts from several tree species were more potent antioxidants than the bark extracts or pure lignans and flavonoids. Although most of the antioxidant activity of the extracts can be explained by the predominant compounds, synergism of polyphenols or compounds in minor amounts strongly contribute to the activity [15].

On the other hand, Välimaa et al. [21] did not confirm antimicrobial, antibacterial, or antifungal properties of extract of silver fir (*A. alba*) KW. Similarly, information on the antifungal properties of silver fir KW and BW extractives against *Trametes versicolor*, *Schizophyllum commune* and *Gloeophyllum trabeum*, as well as antimicrobial properties against *Penicillium expynsum* and *Fusarium solani*, is lacking in the literature. These organisms are known frequently to attack physiologically weakened trees, as well as wood in use. Extracts from BW, which are actually residues from logging operations, have also never been included in in vitro tests for antifungal properties. This information is missing.

The objectives of the present study were (a) to investigate the variability of silver fir (*Abies alba* Mill.) extractives in the radial direction, i.e., in the HW and SW from the pith towards the outermost wood, and in the BW at different distances from the stem, (b) to analyze the chemical composition of extracts of KW located in sapwood and of KW located in heartwood. In the second part of the study, the bioactive potential of the KW and BW extracts of silver fir was investigated. Here, the aim was (c) to measure the antifungal and antimicrobial properties against selected wood-decaying fungi and against selected molds, and to measure the antioxidative properties of the extracts.

## 2. Materials and Methods

Extraction solvents (acetone and cyclohexane, HPLC grade) and solvents for chromatographic analysis (methanol and water, HPLC grade), and formic acid (for LC/MS, 99%)) were purchased from Merck (Sigma-Aldrich Chemie, Taufkirchen, Germany), Carlo Erba Reagents (Chaussée du Vexin, France) and from J.T. Baker (Phillipsburg, NJ, USA). Folin–Ciocalteu phenol reagent (2 N), anhydrous sodium carbonate (99%) and gallic acid monohydrate (HPLC assay, ≥99%), gallic acid (certified reference material), l-ascorbic acid (reagent grade), butylated hydroxyanisole (analytical reference material) and 2,2-diphenyl-1-picrylhydrazyl (DPPH) were also provided by Merck (Sigma-Aldrich Chemie, Taufkirchen, Germany). Lariciresinol (purity ≥ 95%) and isolariciresinol (purity ≥ 95%) were kindly provided by our colleagues from Åbo Akademi University, Laboratory of Organic Chemistry (Prof. Dr. Stefan Willför). Other reference compounds used for chromatographic analysis are commercially available. Epichatechin (purity HPLC) ≥ 99%), coumaric acid (purity (HPLC) ≥ 90%), homovanillic acid (purity (HPLC) ≥ 95%), taxifolin (purity (HPLC) ≥ 99%), and ferulic acid (purity (HPLC) ≥ 90%) were from Extrasynthese (Genay, France). The analytical standards secoisolariciresinol (purity (HPLC) ≥ 95%), pinoresinol (purity (HPLC) ≥ 95%), matairesinol (purity (HPLC) ≥ 95%) and quercetin (purity (HPLC) ≥ 95%) were supplied by Merck (Sigma-Aldrich Chemie, Taufkirchen, Germany). The dimethyl sulfoxide (DMSO) in which the extracts were dissolved, and the potato dextrose agar (PDA) nutrient medium for the fungal assay were purchased from Gram-Mol (Zagreb, Croatia) and DIFCO (Fisher Scientific, Franklin Lakes, NJ, USA), respectively. The fungal and mold isolates of *T. versicolor*, *S. commune*, and *G. trabeum*, *P. expynsum,* and *F. solani* originated from the fungal collection of the Biotechnical Faculty, University of Ljubljana. 

Three silver fir trees (*Abies alba* Mill.) originating from the forest of Kočevska Reka were included in the study. The felling of the trees was carried out in mid-December 2018. The sample discs were taken from the upper parts of the fallen tree trunks. The biometric data and sampling heights of the sample discs taken are given in Table 1.

The disks were planed, after which the boundary between SW and HW was marked, the growth rings counted, their diameters measured, and the sampling sites marked. Various woody tissues were isolated from the stem, knots, and branches, as shown in Figure 1. In the case of stem tissues, more samples of SW and HW were taken in a radial direction, i.e., from the peripheral part towards the pith (Figure 1a).

KW from dead and living branches, i.e., dead knots (DK) and living knots (LK), were also sampled. KW samples were separated into parts embedded in sapwood (LK-SW and DK-SW) and parts embedded in heartwood (LK-HW and DK-HW) (Figure 1b). Samples from living branches were taken at three locations, i.e., where the branch enters the stem, and then twice every 10 cm (Figure 1b). The samples were dried at 40 °C for 24 h and then disintegrated on a Retsch SM 2000 cutting mill using a sieve with 1 mm openings. The ground samples were stored in a dark and cool place until the beginning of the chemical analyzes.

### 2.1. Extraction

Sequential extraction was performed in an accelerated solvent extraction system ASE 350 (Thermo Scientific Dionex). Lipophilic extractives were extracted first with cyclohexane followed by hydrophilic extractives with an acetone/water (95:5, *v*/*v*) mixture. Extraction with each solvent was performed at 100 °C and 103.42 bar (2 × 5 min static cycles) [2]. Prior to extraction, all samples were lyophilized in a Telstar LyoQuest CC1930 freeze dryer for 24 h at −85 °C and 0.045 mbar. Total lipophilic and hydrophilic extractives were determined gravimetrically. Aliquots of the extracts were dried to constant weight. Results were expressed in milligrams of extractives per gram of dry wood (mg/g dw) [22].

### 2.2. Spectrophotometric Analysis

Total phenolics were measured according to the protocol described previously [22,23]. Diluted Folin–Ciocalteu phenol reagent (aq) and an aqueous solution of sodium carbonate (75 g/L) were added to each wood extract. After incubation of the reaction mixtures, absorbance was measured at 765 nm using a Lambada (Perkin-Elmer) UV/Vis spectrophotometer. The results were determined using the standard curve of gallic acid and expressed in milligrams of gallic acid equivalents per gram of dried wood sample (mg GAE/g).

### 2.3. Chromatographic Analysis

#### 2.3.1. Thin Layer Chromatography

Concentrated acetone extracts of LK and DK as well as standards were punctually applied to Merck TLC-plates Silica Gel 60 F254 aluminum sheets (20 cm × 20 cm, Merck KGaA, Darmstadt, Germany) using a micropipette. A mixture of chloroform and ethanol (90:10, *v/v*) was used as the developing solvent system-eluent. Analysis was performed in a saturated chromatographic chamber. For visualization of the separated spots, spraying with sulfuric acid:ethanol in a 50:50 (*v/v*) ratio was used. The migration distance was 8 cm. The compounds were identified based on comparison of the measured retention factors of sample spots with standards. The color of the spots was also considered.

#### 2.3.2. Reversed-Phase High-Performance Liquid Chromatography (RP-HPLC)

Chromatographic analysis was performed on a Thermo Scientific high-performance liquid chromatography (Accela HPLC) system equipped with a photodiode array detector (PDA). Samples were separated on a Thermo Accucore ODS column (4.6 id × 150 mm, 2.6 μm). Water (A) and methanol (B), both containing 0.1% formic acid, served as the mobile phase. The flow rate of the mobile phase was set at 1000 μL/min. The gradient used was 5–95% of solvent (B). The autosampler with sample trays and the column oven were thermostatted to 5 °C and 30 °C, respectively. Absorbance was measured at 280 nm and UV spectra were recorded from 200 nm to 400 nm. Peak identities were investigated by comparing the retention times and UV spectra of the separated compounds with those of the external analytical standards. The chromatographic method was linear (*R*^2^ ≥ 0.99) in the selected concentration range. Samples were measured in triplicate. Identification of phenolic compounds was performed using external standards [18].

### 2.4. In Vitro Antifungal and Antimicrobial Assay

The antifungal and antimicrobial potential of silver fir (*A. alba*) wood extracts was evaluated using the agar well diffusion method [24]. Extracts of LK, DK and BW dissolved in dimethyl sulfoxide (DMSO, 1% and 5% (*v/v*)) were tested. Potato dextrose agar (PDA) was used as the nutrient medium. Three wells were drilled into the medium, with the center of each well located 10 mm from the edge of the Petri dish and then 100 µL of pure DMSO (control), 1% of extract solution, and 5% of extract solution were dropped into the wells. The Petri dishes were then inoculated with selected fungi (*T. versicolor*, *S. commune,* and *G. trabeum*) and molds (*P. expynsum* and *F. solani*) with inoculum or spore suspensions placed in the center of the Petri dish. Petri dishes were incubated in a growth chamber (T = 25 °C, 75% relative humidity). The extracts were tested in six repetitions for each of the tested fungi and molds. The growth of the test organisms was monitored every 2–3 days until the organism grew in one direction to the edge of the petri dish, or until the growth of the fungus ceased. Test results were reported as percentage inhibition of fungal growth in the radial direction (%) [24].

### 2.5. Determination of Antioxidant Potential by the DPPH Method

Antioxidant potential was determined by the 2,2-diphenyl-1-picrylhydrazyl (DPPH) radical scavenging method as already described [8,18,25]. Pure methanol was used as a blank sample. Sample and reference solutions were prepared at five concentrations (1000 mg/L, 500 mg/L, 250 mg/L, 100 mg/L, and 50 mg/L). All the prepared solutions and control were applied into cuvettes (90 µL) and methanolic DPPH reagent solution (2.25 mL) was added. After incubation at room temperature for 30 min, the absorbance of the reagent mixture was measured at 517 nm using a Perkin-Elmer Lambda UV-Vis spectrophotometer.

### 2.6. Statistics

The statistical significance of measured differences was analyzed with basic statistical analysis using a Statgraphics software. The results were as first checked for normal distribution, and analysis of variance (ANOVA) and Fisher’s least significant difference (LSD) procedure at a 95.0% confidence level were performed. The structural formulas of lignans were drawn by using a Perkin Elmer’s ChemDraw 20.1 software.

## 3. Results and Discussion

### 3.1. Contents of Lipophilic and Hydrophilic Extractives and Total Phenols

Figure 2 provides information on the average content of hydrophilic extractives (HE), lipophilic extractives (LE) and total phenols (TP) in sapwood (SW), heartwood (HW), and wood of living knots (LK), wood of branches (BW), and dead knots (DK).

All tissues investigated contained significantly larger amounts of HE than LE (Figure 2). The largest amounts of HE and LE were measured in DK samples (210.4 mg/g HE and 20.1 mg/g LE), while SW yielded the lowest amounts of extractives (14.8 mg/g HE and 4.7 mg/g LE). These results are in good agreement with the report of a Finnish research group, which found 0.70% of HE in SW, 2.1% of HE in HW, 13% of HE in LK, and 15% of HE in DK [2] (Figure 2).

As can be seen from Figure 2, knots contain the highest amounts of HE, LE, and TP. LK samples contained an average of 162.6 milligrams of extractives per gram of absolutely dry wood, and DK samples contained 230.5 mg/g. BW samples contained only slightly less extractives than the knots (158.0 mg/g). HW (39.3 mg/g) and SW (19.5 mg/g) contained significantly less extractives. HW contained on average about five times lower amounts of total extractives than knots. The same trend was observed in the content of TP. The content of TP was lower than the content of total extractives in all woody tissues. The SW with the lowest phenolic content had an average of only 1.3 milligrams of TP per gram of dry wood; larger amounts were measured in HW samples. BW and the knots were found to be rich sources of phenolic compounds.

### 3.2. Chemical Identities of Phenolic Extractives of Silver Fir Wood

The chemical structures of lignans and the developed TLC plate of standards and extracts of silver fir (*A. alba*) KW are shown in Figure 3 and Figure 4, respectively. On the TLC plates, at least fourteen different spots of silver fir extracts were observed. Based on the existing literature [26], spot colors and a chromatogram of the standards on the TLC plate, the following lignans were identified in the extracts of LK and DK: isolariciresinol, secoisolariciresinol, lariciresinol, pinoresinol, and matairesinol (Figure 3 and Figure 4, Table 2). Isolariciresinol and secoisolariciresinol are the most polar compounds among the identified lignans, with four hydroxyl groups, and exhibit the lowest RF. Secoisolariciresinol shows a little bit higher RF due to a higher hydrodynamic volume of the molecule. The third identified spot belongs to lariciresinol, with three hydroxyl groups. Highest on the TLC chromatogram, however, we identified matairesinol and pinoresinol. Matairesinol and pinoresinol have two hydroxyl groups and have the most non-polar character among the mentioned compounds, so their RF is also the highest. In Table 2, the retention factors of the spots on the TLC chromatograms of extracts and standards are listed.

The retention factors, spot colors and a chromatogram from the existing literature were used for identification [26]. Small differences in retention factors between the standards and the compounds in the extract are due to the interaction of the compounds with each other, as well as interactions with the stationary, mobile phase, and the atmosphere in the developing chamber. In addition to the identified compounds, other compounds were detected in the TLC analysis (Figure 4) that were also not detected in the HPLC analysis. The identification of these compounds/spots remains the goal of our future research activities.

The results of spectrophotometric and chromatographic analysis of the wood extracts of silver fir (*A. alba*) tissues showed significant variability in the content of phenolic extractives in the studied samples (Figure 5, Table 3). The highest peaks in the HPLC traces of silver fir wood extracts were attributed to lignans (Figure 5), which have already been confirmed as the dominant group of phenols in silver fir wood extracts [2,8]. DK also contained phenolic acids and flavonoids, represented by the lower peaks (Figure 5a). As the most polar compound, the flavonoid epicatechin (Epi, tr = 8.0 min) eluted first, followed by the phenolic acids homovanillic acid (HVA, tr = 8.1 min) and coumaric acid (CA, tr = 9.5 min), the flavonoid taxifolin (Tax, tr = 9.7 min), ferulic acid (Fer, tr = 10.0), the lignans isolariciresinol (Iso-Lari, tr = 10.2 min), lariciresinol (Lari, tr = 11.5 min), secoisolariciresinol (Seco, tr = 11.7 min), pinoresinol (Pino, tr = 12.6 min) and matairesinol (Matai, tr = 12.9 min), and the less polar flavonoid quercetin was eluted from the column last (Qur, tr = 13.6 min) (Figure 5).

As with the extracts of KW, the highest peaks on the HPLC chromatogram for BW were represented by lignans (Figure 5b). There is an obvious difference between the chromatograms in Figure 5 in the heights of the peaks, especially those representing lignans (peaks nos. 6, 7, 8, 9, and 10). The content of lignans in the acetone extracts of DK-SW (Figure 5a) was much higher than in BW (Figure 5b) and HW (Figure 5c). The most abundant lignan in the wood of branches was secoisolariciresinol. The results presented in Table 3 show how the content of extractives in BW, especially lignans, decreases with increasing distance from the trunk. The greatest difference between the extracts of the wood of the branch close to the trunk (BW0) and the wood of the branch 10 cm from the trunk (BW10) was in the content of isolariciresinol. The extract of BW 20 cm from the trunk also differed in the content of lariciresionol and secoisolariciresinol. Samples of different aged parts of HW and SW (Figure 1 and Table 3) showed a difference in both the content and chemical identity of the extractives. Younger parts of SW and HW contained more TE and TP than the older parts. On the other hand, the older parts of HW contained higher amounts of lignans. The predominant lignan in HW was matairesionl and this is consistent with the report of Willför et al. [2]. The younger parts of HW also contained more HVA, CA, and Tax than the older parts of HW. Among the knots (LK and DK), the highest contents of the identified extractives were measured in parts that were incorporated into the sapwood (LK-SW and DK-SW) (Figure 1), and significantly less extractives were contained in the parts of the knots incorporated into the heartwood (LK-HW and DK-HW). LK samples contained the highest amount of secoisolariciresinol in both SW and HW. DK-SW samples contained the largest amounts of isolariciresinol, while DK-HW contained the highest amounts of lariciresionol.

### 3.3. Antifungal and Antioxidant Properties of Knotwood and Branchwood Extracts of Silver Fir

#### 3.3.1. In Vitro Growth Inhibition of Selected Fungi and Molds by Extractives from Knotwood and Branchwood of Silver Fir

The in vitro fungal assay showed that extracts of KW and BW of silver fir (*A. alba*) inhibited mycelium growth of the test fungi (Figure 6). On average, inhibition of fungal growth was just below or above 20% for the 5% solutions containing extracts of KW and BW. In contrast, little or no inhibitory properties were measured for the 1% solutions. The white-rot fungi (*Trametes versicolor* and *Schizophyllum commune*) and the brown-rot fungus (*Gloeophyllum trabeum*) were inhibited by the 5% solution of KW and BW extract. 

Extracts of KW and BW of silver fir also inhibited mold growth (Figure 7). On average, the inhibition was slightly higher than with the fungi (Figure 6). If we disregard the low inhibitory effect of the 5% extract of KW on *Fusarium solani*, growth inhibition by the 5% extract solutions was about 25%. The 1% solutions of KW and BW extracts exhibited about 10% growth inhibition of the tested molds. Concentrated KW solutions showed the highest growth inhibition for *Penicillium expynsum*, while 5% BW solutions inhibited the growth of *Fusarium solani*. The hydrophilic extractives of silver fir woody tissue showed relatively low but a relevant inhibitory effect on the growth of fungi and molds. Our results tend to complement the findings of Välimaa et al. [21], who reported a low antimicrobial and antifungal effect of silver fir KW extractives against gram-positive pathogenic bacteria *Bacillus cereus* and *Listeria monocytogenes*, and against the yeast *Candida albicans*.

#### 3.3.2. Antioxidant Potential of Silver Fir Knotwood and Branchwood Extracts

Figure 8 shows the DDPH free radical scavenging activity (RSA) for hydrophilic extracts of HW, DK, LK, and BW of silver fir (*Abies alba*) and for the selected references (gallic—GaA and ascorbic acid—AsA). RSA increased with increasing concentration of control antioxidants or extracts.

The lowest DPPH RSA was measured for HW (Figure 8). The extracts of LK, DK, and BW showed similar RSA, which were significantly higher than the RSA of HW extracts. At a concentration of 100 mg/L, the wood extracts of silver fir showed an average RSA that was 60% lower than GaA and 30% lower than AsA. At a concentration of 500 mg/L, the difference was only about 20% compared to both references. As the concentration of silver fir wood extracts increased, the antioxidant activity improved significantly. At the highest concentration, the RSA of the KW and BW extracts was only slightly lower than the RSA of the reference compounds (Figure 8). Wood extractives of black locust with antioxidant properties and without inhibitory activity against the wood decay fungi measured in vitro have already been shown to inhibit fungal decay of wood [18]. The KW and BW extractives of silver fir, with their antioxidant effects, may have an important influence on the natural durability of these tissues. The sites on the tree where a branch enters the stem can be understood as sites exposed to mechanical injury and attack by harmful organisms. High contents of extractives in woody tissues at the base of branches could provide constitutive protection. However, these statements need to be supplemented by the results of further laboratory and field tests. The results of the present investigation describe the woody tissues of silver fir as a potential source of natural antioxidants. Before the extractives as such can be used as agents in functional food and/or dietary supplements, the unidentified compounds should be chemically characterized and their biological activity tested. 

## 4. Conclusions

Our study showed that stemwood of silver fir (*Abies alba* Mill.) can contain five times higher amounts of hydrophilic than lipophilic extractives, and this difference is even greater in the KW samples. Lignans, phenolic acids, and flavonoids were identified in the wood extracts of silver fir. The younger parts of the SW and HW contained larger amounts of phenolic extractives, while the older parts of HW contained higher amounts of identified lignans. The results of the investigation show that the largest amounts of identified extractives were measured in the KW taken from the sapwood; significantly lower amounts were found in parts of the KW taken from the heartwood. LK, DK from both sapwood and heartwood, as well as BW samples, contained the highest amount of secoisolariciresinol. Isolariciresinol and lariciresinol were the characteristic compounds for sections of DK embedded in sapwood and heartwood, respectively. The content of extractives in BW was highest in samples taken at the point where a branch enters a stem. In vitro antifungal assay demonstrated that the KW and BW extracts of silver fir clearly inhibited the growth of the wood decaying fungi and the molds tested. The free radical scavenging activity shown by KW and BW extractives was higher than the antioxidant activity of HW extractives. It was demonstrated that, in addition to KW and HW, the BW of silver fir also contains hydrophilic extractives with antifungal and antioxidant properties. Knots from peripheral wood of a silver fir stem and wood of living branches, i.e., tissues rich in phenolic compounds, show high biorefinery potential.

## Figures and Tables

**Figure 1 molecules-26-06412-f001:**
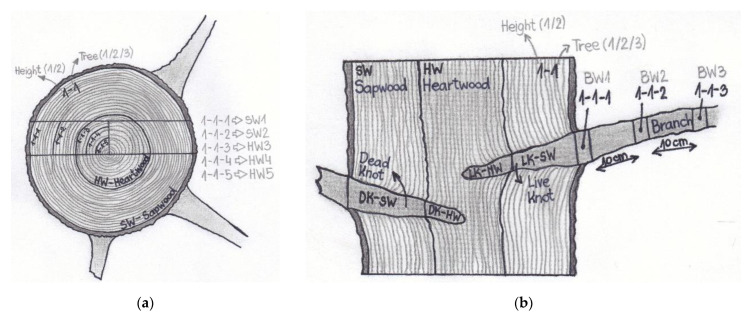
Simple scheme showing the sampling sites for stemwood samples (**a**), and for knotwood and branchwood samples (**b**) of silver fir (*A. alba*).

**Figure 2 molecules-26-06412-f002:**
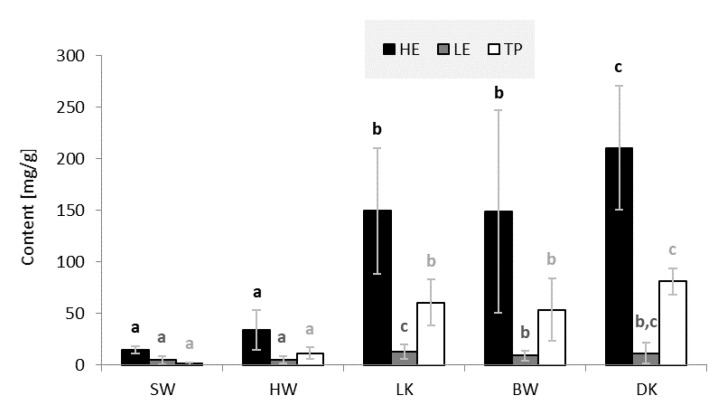
Content of and hydrophilic extractives (HE, acetone-soluble compounds), lipophilic extractives (LE, cyclohexane-soluble compounds) and total phenols (TP) in sapwood (SW), heartwood (HW), wood of living knots (LK), wood of branches (BW) and wood of dead knots (DK) of silver fir (*A. alba*). a–c Different letters on the error bars of the same series of columns indicate statistically significant differences at a 95% confidence level (LSD test).

**Figure 3 molecules-26-06412-f003:**
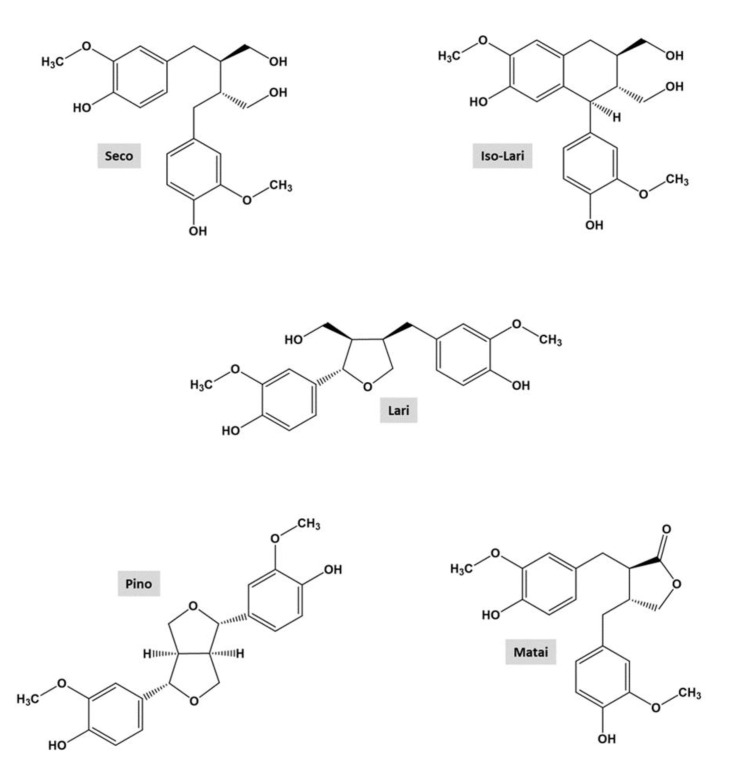
Chemical structures of the lignans in knotwood of silver fir (*Abies alba* Mill): secoisolariciresinol (Seco), isolariciresinol (Iso-Lari), lariciresinol (Lari), pinoresinol (Pino), matairesinol (Matai).

**Figure 4 molecules-26-06412-f004:**
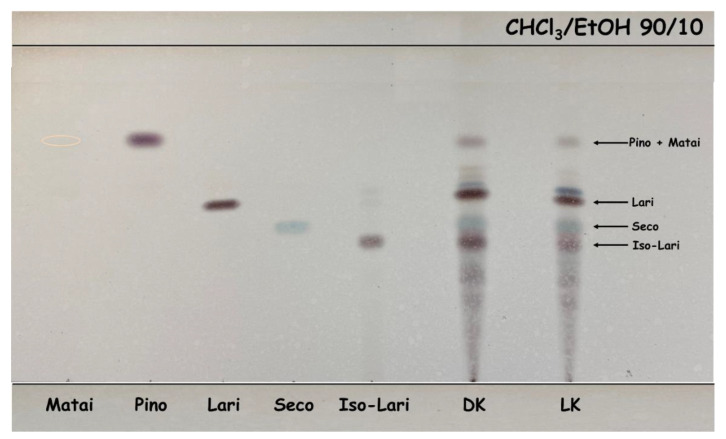
Developed TLC-plates of standards (left) and extracts of silver fir (*A. alba*) knots (right). matairesinol (Matai), pinoresinol (Pino), isolariciresinol (Iso-Lari), lariciresinol (Lari), secoisolariciresinol (Seco). Living knots (LK) and dead knots (DK).

**Figure 5 molecules-26-06412-f005:**
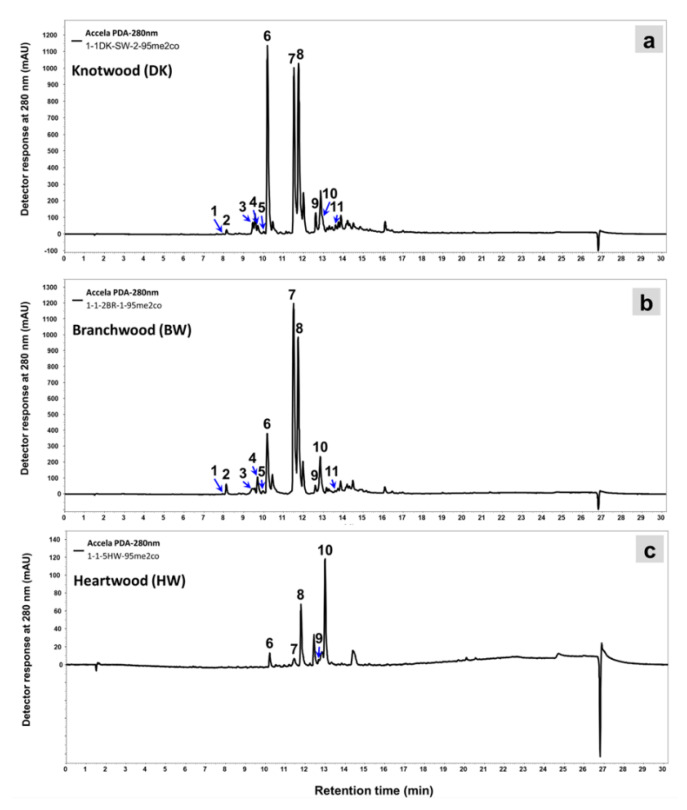
HPLC–PDA chromatograms of acetone extracts of silver fir (*A. alba*) measured at 280 nm. (**a**) Wood of a dead knot taken from the sapwood (DK–SW). (**b**) Wood of a branch. The sample was taken at a distance of 10 cm from the stem (BW10). (**c**) The oldest, innermost part of heartwood (HW5) (Figure 1). For the identity of the chromatographic peaks, see Table 3.

**Figure 6 molecules-26-06412-f006:**
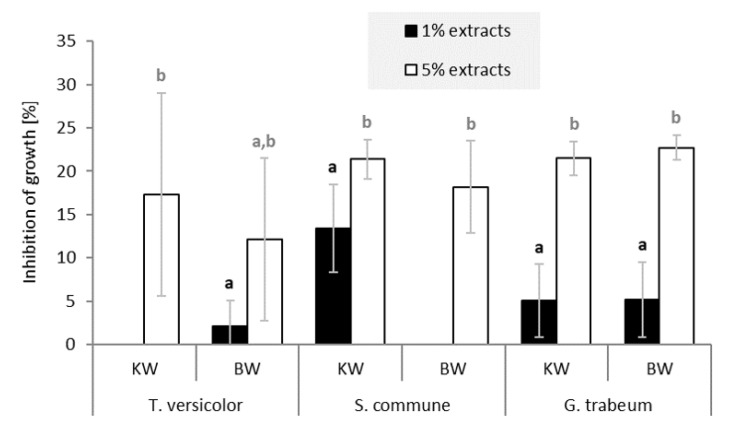
Growth inhibition of the white-rot fungi *Trametes versicolor* and *Schizophyllum commune*, and the brown-rot fungus *Gloeophyllum trabeum* by crude hydrophilic extracts of silver fir (*Abies alba*). The columns represent the average inhibition achieved with 1% and 5% solutions (DMSO, *w/v*) of knotwood (KW) and branchwood (BW) extracts. a,b Different letters on the error bars of the same series of columns indicate statistically significant differences at a 95% confidence level (LSD test).

**Figure 7 molecules-26-06412-f007:**
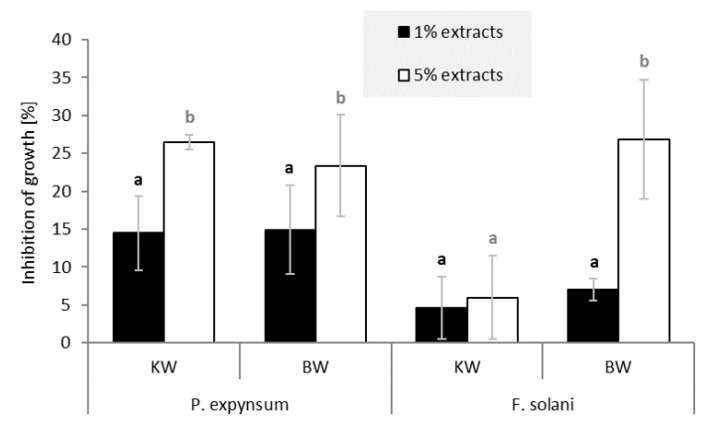
Growth inhibition of molds *Penicillium expynsum* and *Fusarium solani* by crude hydrophilic extracts of silver fir (*Abies alba*). The columns represent the average inhibition achieved with 1% and 5% solutions (DMSO, *w/v*) of knotwood (KW) and branchwood (BW) extracts. a,b Different letters on the error bars of the same series of columns indicate statistically significant differences at a 95% confidence level (LSD test).

**Figure 8 molecules-26-06412-f008:**
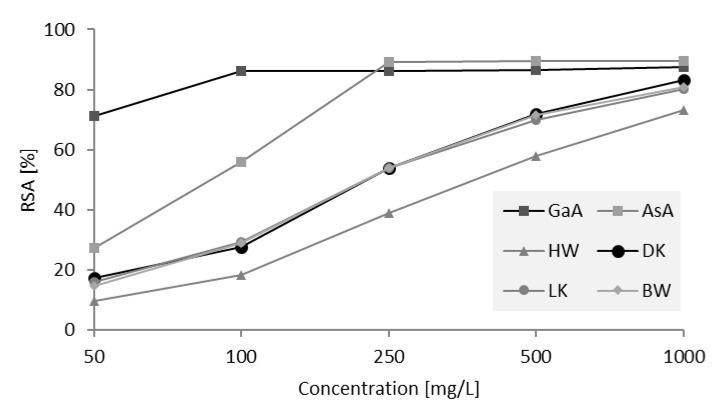
Radical scavenging activity (RSA, %) for reference antioxidants (gallic (GaA) and ascorbic acid (AsA)) and for hydrophilic extracts of heartwood (HW), knotwood of dead branches (dead knot, DK), knotwood of living branches (living knot, LK) and branchwood (BW) of silver fir (*A. alba*). The reference and the extract solutions were tested at concentrations of 50 mg/L, 100 mg/L, 250 mg/L, 500 mg/L, and 1000 mg/L.

**Table 1 molecules-26-06412-t001:** Information on sample discs of silver fir (*A. alba*).

Sample Disc	Sampling Height (m)	Tree Diameter (with Bark) (cm)	Heartwood Diameter (cm)	Age	Age of Heartwood
1-1	12	28.90	18.95	64	38
2-1	16	23.75	13.2	55	27
1-2	7	39.15	22.3	61	39
2-2	16	28.10	13.9	39	21
1-3	13	40.70	28.75	62	47
2-3	21	25.75	9.9	31	14

**Table 2 molecules-26-06412-t002:** Retention factors of spots on TLC chromatograms of silver fir (*A. alba*) knots extracts; compounds represented by spots and retention factors of standards.

Spot	Average Retention Factor (RF)	Compound	Retention Factor of Standards (RF)
A	0.41	Isolariciresinol	0.42
B	0.46	Secoisolariciresinol	0.46
C	0.55	Lariciresinol	0.53
D	0.70	Matairesinol	0.70
E	0.71	Pinoresinol	0.72

**Table 3 molecules-26-06412-t003:** Contents of total extractives (TE), total phenols (TP) and the identified phenolic compounds in samples of sapwood (SW), heartwood (HW), knotwood (LK and DK), and branchwood (BW) of silver fir (*A. alba*). The results in the table are presented as average values, with standard deviations (in brackets).

		SW		HW			LK		BW			DK	
No.		1	2	3	4	5	HW	SW	0	10	20	HW	SW
	TE [mg/g]	20.4 (5.4)	18.8 (7.6)	49.0 (21.9)	37.6 (22.4)	31.4 (16.7)	128.3 (40.3)	196.9 (51.7)	258.9 (71.7)	151.4 (97.3)	74.8 (33.3)	179.3 (44.0)	281.7 (25.5)
	TP [mg/g]	1.7 (1.7)	0.9 (1.2)	13.8 (5.6)	11.2 (6.4)	10.0 (4.0)	50.7 (18.8)	70.4 (21.7)	79.8 (21.0)	54.8 (29.3)	29.0 (13.8)	56.4 (37.7)	86.3 (11.0)
1	Epi [mg/g]	0.1 (0.0)	* ND (0.0)	* ND (0.0)	* ND (0.0)	* ND (0.0)	* ND (0.0)	0.1 (0.0)	0.1 (0.0)	0.1 (0.0)	0.1 (0.0)	* ND (0.0)	0.1 (0.0)
2	HVA [mg/g]	* ND (0.0)	* ND (0.0)	0.7 (0.3)	0.3 (0.4)	0.3 (0.2)	1.3 (0.6)	2.2 (0.6)	2.3 (0.9)	1.8 (1.1)	1.0 (0.8)	1.5 (1.1)	1.5 (0.3)
3	CA [mg/g]	* ND (0.0)	* ND (0.0)	0.4 (0.2)	0.2 (0.2)	0.2 (0.2)	0.5 (0.3)	0.3 (0.2)	0.1 (0.1)	0.1 (0.1)	0.2 (0.2)	0.2 (0.3)	0.5 (0.2)
4	Tax [mg/g]	* ND (0.0)	* ND (0.0)	0.2 (0.1)	0.1 (0.1)	0.1 (0.0)	0.5 (0.3)	0.9 (0.3)	1.2 (0.5)	0.8 (0.5)	0.4 (0.2)	0.6 (0.4)	0.5 (0.1)
5	Fer [mg/g]	* ND (0.0)	* ND (0.0)	0.1 (0.1)	0.0 (0.0)	0.0 (0.0)	0.1 (0.1)	0.2 (0.1)	0.2 (0.1)	0.1 (0.1)	0.1 (0.1)	0.1 (0.1)	0.1 (0.0)
6	Iso-Lari [mg/g]	* ND (0.0)	* ND (0.0)	1.0 (1.1)	0.5 (0.3)	0.6 (0.4)	7.2 (6.5)	20.1 (14.4)	24.5 (17.9)	13.9 (11.6)	7.9 (13.9)	15.3 (8.9)	46.9 (35.1)
7	Lari [mg/g]	* ND (0.0)	* ND (0.0)	0.4 (0.4)	0.2 (0.2)	0.5 (0.7)	8.5 (7.9)	26.1 (17.8)	32.3 (20.8)	22.5 (17.5)	11.0 (13.3)	25.4 (15.7)	25.9 (15.2)
8	Seco [mg/g]	0.1 (0.0)	0.1 (0.1)	0.2 (0.1)	0.4 (0.3)	0.7 (0.7)	12.9 (9.3)	29.8 (16.1)	37.6 (22.1)	22.8 (15.8)	11.2 (8.9)	19.3 (10.7)	21.3 (13.1)
9	Pino [mg/g]	0.1 (0.0)	* ND (0.0)	* ND (0.0)	* ND (0.0)	* ND (0.0)	0.7 (0.4)	1.7 (0.9)	2.0 (1.0)	1.1 (0.7)	0.6 (0.7)	1.2 (0.6)	2.5 (1.6)
10	Matai [mg/g]	* ND (0.0)	* ND (0.0)	0.6 (0.5)	1.5 (1.2)	1.6 (1.3)	4.6 (2.6)	7.8 (4.5)	10.0 (5.5)	6.1 (3.9)	3.5 (3.0)	6.7 (4.0)	8.2 (4.9)
11	Qur [mg/g]	* ND (0.0)	* ND (0.0)	* ND (0.0)	* ND (0.0)	* ND (0.0)	0.2 (0.2)	0.4 (0.3)	0.6 (0.3)	0.3 (0.2)	0.1 (0.5)	0.7 (0.4)	1.4 (0.7)

Total extractives (TE), total phenols (TP), epicatechin (Epi), homovanillic acid (HVA), coumaric acid (CA), taxifolin (Tax), ferulic acid (Fer), isolariciresinol (Iso-Lari), lariciresinol (Lari), secoisolariciresinol (Seco), pinoresinol (Pino), matairesinol (Matai), and quercetin (Qur). Peripheral part of the sapwood (SW1) and inner part of the sapwood (SW2). Outermost heartwood (HW3), intermediate heartwood (HW4) and innermost heartwood (HW5). Wood of living branch (living knot) in heartwood (LK-HW), wood of living branch (living knot) in sapwood (LK-SW). Branchwood sample right next to the stem (BW0), branchwood sample that was taken 10 cm from the site where the branch enters the stem (BW10), and branchwood sample taken 20 cm from the site where branch enters the stem (BW20). Wood of a dead branch (dead knot) in heartwood (DK-HW), wood of a dead branch (dead knot) in sapwood (DK-SW). * ND–non-detected.

## Data Availability

The data presented in this study are available on request from the corresponding authors.

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
