# Peer review of "Wood Extractives of Silver Fir and Their Antioxidant and Antifungal Properties"

_molecules, 2021, doi:10.3390/molecules26216412_

Round 1
Reviewer 1 Report
In this manuscript entitled “Wood extractives of silver fir and their antioxidant and antifungal properties” the authors provided a detailed and clear description of the work in every part, showing that stemwood of silver fir (Abies alba Mill.) can contain five times higher amounts of hydrophilic than lipophilic extractives, and this difference is even greater in the knot wood samples. Moreover, it was demonstrated that, in addition to knotwood and heartwood, the branchwood of silver fir also contains hydrophilic extractives with antifungal and antioxidant properties.
The experimental part is well described, and the discussion is well supported by experimental data. However, manuscript needs minor changes before publications:
-The abstract and conclusion, although extensive, are very generic, they should be supported with some of the results obtained.
-Please use acronyms for sapwood, heartwood, knotwood and branchwood starting from the introduction and use them along the overall text for easier reading.
-Please revise the term “extractives” … if possible, is better to replace it with “extracts” in the overall manuscript.
Reviewer 2 Report
In the ms submitted, the authors have analysed wood extractives in the knotwood of silver fir (Abies alba). As a comparison, extracts of heartwood, sapwood and branchwood were also analysed. Gravimetric method, together with TLC and HPLC were used. In addition, the effects of extracts in growth inhibition tests against some wood-decaying fungi and molds were studied, as well as the antioxidant properties.
The composition of the extractives in the knotwood and heartwood of silver fir have been earlier studied, but the present study has focussed on the radial distribution of the extractives in these tissues. Furthermore, no information about extractives in branchwood existed earlier.
The results show that knotwood and branchwood are rich sources of secondary metabolites, e.g. lignans, flavonoids and phenolic acids, with highest concentration there where the knotwood exist inside the sapwood, and where the branch enters the stem. The extracts have antifungal and antioxidant properties.
I have following comments:
Extractives were analysed with thin-layer chromatography followed by spraying with sulphuric acid -ethanol mixture. The Rf values of the sample spots and their colour were compared with those of the external pure reference compounds. The Rf values of some main spots did not match with those of the external standards, and some reasons for those were discussed. To me it looks like the loading spot has been overloaded with the concentrated extract, and that has affected the run. I recommend you to 1) dilute your extract before sample pipetting. If the spots are too faint to see after spraying, 2) pipette the extract in a short streak instead of a spot, so that the loading streak is not overloaded. External reference compounds should be run in the same TLC plate to ensure that the running conditions are exactly the same. In addition to normal samples, you can spike the extract sample with the external reference compound (one compound per loading). Then the corresponding spot will show bigger after spraying, showing that the compound really migrates in the spot of interest. My opinion is that these kind of further assays need to be done before the ms is accepted for publication.
Fig. 1 is a very good figure and shows the sampling points. Could Fig. 1a be bigger so that the text is easily visible for those who prefer to read from paper copies.
In figures and their corresponding legends, please, use the same order for presentation of the bars/structures and introducing them in the legend, for example in Fig. 2, Fig. 3. There are no LK and DK in Fig. 3 so these can be omitted from the legend.
From line 321 forward: please, do not use too many abbreviations since reading comes difficult. For example, the compounds can be written in full in the main text.
Lines 389-391: According to Fig. 6, both 5% KW and BW extract inhibited the growth of G. trabeum, and also both inhibited the growth of T. versicolor and S. commune.
Lines 432-433: There exist several unidentified compounds in the extracts. Use of them as such in functional food or dietary supplements requires much advanced characterisation and further tests.
Minor comments:
It would be good to have Table 3 in one page (without any page break like now).
Table 3: coumaric acid instead of coumarillic acid (CA)
Author Response
We would like to thank the Reviewer 2 for comments and useful suggestions. We implemented practically all of them and our response is point by point provided below.
I have following comments:
Extractives were analysed with thin-layer chromatography followed by spraying with sulphuric acid -ethanol mixture. The Rf values of the sample spots and their colour were compared with those of the external pure reference compounds. The Rf values of some main spots did not match with those of the external standards, and some reasons for those were discussed. To me it looks like the loading spot has been overloaded with the concentrated extract, and that has affected the run. I recommend you to 1) dilute your extract before sample pipetting. If the spots are too faint to see after spraying, 2) pipette the extract in a short streak instead of a spot, so that the loading streak is not overloaded. External reference compounds should be run in the same TLC plate to ensure that the running conditions are exactly the same. In addition to normal samples, you can spike the extract sample with the external reference compound (one compound per loading). Then the corresponding spot will show bigger after spraying, showing that the compound really migrates in the spot of interest. My opinion is that these kind of further assays need to be done before the ms is accepted for publication.
Response: TLC experiment was repeated as suggested. Figure 4 was replaced and RF in Table 2 were corrected
Fig. 1 is a very good figure and shows the sampling points. Could Fig. 1a be bigger so that the text is easily visible for those who prefer to read from paper copies.
Response: We have enlarged it as we could. We assume that figure 1 can be enlarged in digital version of the manuscript as well.
In figures and their corresponding legends, please, use the same order for presentation of the bars/structures and introducing them in the legend, for example in Fig. 2, Fig. 3. There are no LK and DK in Fig. 3 so these can be omitted from the legend.
Response: Figure labels were corrected according to R2 suggestion
From line 321 forward: please, do not use too many abbreviations since reading comes difficult. For example, the compounds can be written in full in the main text.
Response: We have changed abbreviations of the compounds in their full names. At this point we would like to mention that another reviewer (R1) suggested to replace names of tissues (heartwood, sapwood, etc) with their corresponding abbreviations allover the manuscript, which we did. We have decided to keep figure and table labels as they were.
Lines 389-391: According to Fig. 6, both 5% KW and BW extract inhibited the growth of G. trabeum, and also both inhibited the growth of T. versicolor and S. commune.
Lines 432-433: There exist several unidentified compounds in the extracts. Use of them as such in functional food or dietary supplements requires much advanced characterisation and further tests.
Response: We agree with this comment and have changed the statement. It sounds now “Before the extractives as such can be used as agents in functional food and/or dietary supplements, the unidentified compounds should be chemically characterized and their biological activity tested.«
Minor comments:
It would be good to have Table 3 in one page (without any page break like now).
Response: Table 3should be in one page now
Table 3: coumaric acid instead of coumarillic acid (CA)
Response: Corrected
Reviewer 3 Report
The subject of the research undertaken is interesting, but not the latest. There are already several studies on the qualitative and quantitative composition that included lignans, phenolic acids and flavonoids. Nevertheless, the originality of this experiment. Undoubtedly, an interesting issue is the microbiological activity of extracts from various parts of fir.
However, my comments concern:
The abstract is extensive, but does not contain essential information. Please detail the obtained results. Report ranges and specific components that have been identified.
Keywords: there are too many of them, the authors are asked to indicate the most important keys for this study.
The introduction should be shortened with redundant information, and supplemented with data introducing the causal relationship and purposefulness of the research undertaken. Please indicate the identity of the results obtained in relation to the actual state of knowledge in this field.
The methodology lacks data on the statistical analysis procedure. It is not clear on what basis the statistical differences shown in the figures (Figs. 2, 6 and 7) are indicated. Authors are kindly requested to complete this deficiency.
Author Response
Response:
We would like to thank the Reviewer 3 for comments and useful suggestions. We implemented practically all of them and our response provided below.
Abstract. We have corrected the Abstracts and have added selected results, as well, we have included compounds being identified.
Key words: the list is shortened
Introduction: Redundant information is omitted and we think that the gaps in current knowledge on silver fir extractives became well identified now. As well, we think that the aim of the work defines the intention of the study in relation to the missing information on silver fir extractives, which was provided by reviewing the literature.
Material and Methods: We have overlooked in the final version of the manuscript that we lost subchapter on statistical analysis when merging the documents. It is added now.
Reviewer 4 Report
Dear Authors,
Your manuscript described the antifungal and antioxidant properties from extracts of Abies alba (silver fir tree) additionally of some chemical characterization. Study was adequately presented as well the results. The only concern lies on the extremely traditional methods that could have been updated.
Author Response
Response: We would like to tank R4 for the assessment of the manuscript. We can agree with the comment that somehow traditional methods were used in this work, but hopefully, the helped to reveal new information on extractives of silver fir wood and bark.
Round 2
Reviewer 4 Report
Dear Authors,
Thank you for addressing the question.
Author Response
Dear reviewer, dear Editors,
In the name of authors I apologize for overlooking this comment.
We have corrected the statement (in new version of the manuscript) in L367-369. It sounds now "The white-rot fungi (Trametes versicolor and Schizophyllum commune) and the brown-rot fungus (Gloeophyllum trabeum) were inhibited by the 5 % solution of KW and BW extract. "
"cumaric" was changed with "coumaric" allover the text.
All changes are done with Track changes
Bet regards
Primož
